# Identification and Characterization of Abiotic Stress Responsive CBL-CIPK Family Genes in *Medicago*

**DOI:** 10.3390/ijms22094634

**Published:** 2021-04-28

**Authors:** Wenxuan Du, Junfeng Yang, Lin Ma, Qian Su, Yongzhen Pang

**Affiliations:** 1Institute of Animal Science, Chinese Academy of Agricultural Sciences, Beijing 100193, China; n053727@163.com (W.D.); fyang63@ibcas.ac.cn (J.Y.); malin@caas.cn (L.M.); suqian2021425@163.com (Q.S.); 2Key Laboratory of Plant Resources and Beijing Botanical Garden, Institute of Botany, Chinese Academy of Sciences, Beijing 100093, China; 3Key Laboratory of Forage and Endemic Crop Biotechnology, Ministry of Education, School of Life Sciences, Inner Mongolia University, Hohhot 010010, China

**Keywords:** *Medicago sativa*, *Medicago truncatula*, *CBL* genes, *CIPK* genes, abiotic stresses, ABA, defense compounds, expression profiling

## Abstract

The calcineurin B-like protein (CBL) and CBL-interacting protein kinase (CIPK) play important roles in plant signal transduction and response to abiotic stress. Plants of *Medicago* genus contain many important forages, and their growth is often affected by a variety of abiotic stresses. However, studies on the CBL and CIPK family member and their function are rare in *Medicago*. In this study, a total of 23 *CBL* and 58 *CIPK* genes were identified from the genome of *Medicago sativa* as an important forage crop, and *Medicaog truncatula* as the model plant. Phylogenetic analysis suggested that these *CBL* and *CIPK* genes could be classified into five and seven groups, respectively. Moreover, these genes/proteins showed diverse exon-intron organizations, architectures of conserved protein motifs. Many stress-related *cis*-acting elements were found in their promoter region. In addition, transcriptional analyses showed that these *CBL* and *CIPK* genes exhibited distinct expression patterns in various tissues, and in response to drought, salt, and abscisic acid treatments. In particular, the expression levels of *MtCIPK2* (*MsCIPK3*), *MtCIPK17* (*MsCIPK11*), and *MtCIPK18* (*MsCIPK12*) were significantly increased under PEG, NaCl, and ABA treatments. Collectively, our study suggested that *CBL* and *CIPK* genes play crucial roles in response to various abiotic stresses in *Medicago*.

## 1. Introduction

Plants are exposed to various adverse environment stresses during their growth and development process, including high salinity, drought, cold, and pathogens. Plants have evolved complex signal transduction pathways to cope with these fluctuating environments during their life cycle [1]. Calcium, an important second messenger in plant cell signaling, involves various signal transduction pathways [2]. Calcium is able to convert external signals into cytoplasmic information to further drive the response to specific stimuli [3]. Calcium signals are mainly perceived by Ca^2+^ sensors, including calmodulins (CAMs), calmodulin-like proteins (CMLs), calcium-dependent protein kinases (CDPKs), and the plant-specific calcineurin B-like proteins (CBLs) [4]. The CBLs belong to a unique group of calcium sensors in plants that can be specifically targeted by CBL-interacting protein kinases (CIPKs) to transduce the perceived calcium signal upon various stimulations. CBLs and CIPKs together form the Ca^2+^-mediated CBL-CIPK network [5].

CBL proteins share some signature motifs or domains. CBL proteins contain an important structural component as calcium binding sites, which can capture Ca^2+^ ions through four EF hand domains, and the number of amino acids spaced between the four EF-hand domains is conservative [6,7]. Furthermore, serine residue in the PFPF motif, is necessary for CIPKs to phosphorylate CBLs [8]. There are two important domains in the C-terminal. The conserved NAF motif is responsible for the interaction with Ca^2+^ bound CBLs, thus activating targeted CIPKs [9,10]. Another important domain is the protein phosphotase interaction (PPI) domain, which can target the special member of the protein phosphotase 2C [11].

*CBL* and *CIPK* genes have been identified in many plant species, including *Arabidopsis thaliana* [2], *Oryza sativa* [12], *Brassica napus* [1], and *Ananas comosus* [13]. Many studies have demonstrated that the CBL and CIPK proteins can function properly. For example, the over-expression of *AtCBL5* confer salt tolerance in the transgenic *Arabidopsis* plants [14]. AtCIPK8 itself can regulate the low-affinity phase of the nitrate primary response [15]. Meanwhile, the CBL and CIPK can work together as a network to widely regulate various metabolism processes in plant response to abiotic stress and nutrient signaling cascades [16,17,18], which included, but were not limited to, the transport of sodium (Na^+^), potassium (K^+^), nitrate (NO^3−^) cross plasma membrane and vacuole membrane, and in auxin and ABA signal transduction [19]. Meanwhile, the CBL-CIPK network can transfer the kinase to the plasma membrane and activate the plasma membrane-localized Na^+^/H^+^ antiporter AtNHX7 (SOS1) and the vacuolar H^+^-ATPase to improve salt tolerance in roots of *Arabidopsis* (*AtCBL4*-*AtCIPK24* pathway) [3,20,21]. Moreover, *AtCBL10* can target *AtCIPK24*, and this interaction is related to the vacuolar compartments and can protect shoots from salt stress [22].

Alfalfa (*M. sativa*) is a perennial legume plant, which is one of the important forage crops. Alfalfa had high yield (representing about 2600 kg per hectare) and high protein content (with about 15–20% of crude protein) [23,24]. However, the genetic analysis on the tetraploid alfalfa genome is complicated due to the general intolerance of inbreeding. On the other hand, alfalfa is not tolerant to many abiotic stresses in the ecosystem [24]. *M. truncatula* emerged as model legumes for fundamental research in the 1990s [25]. As a reference species, *M. truncatula* has many advantages, such as short life cycle, easy control of diploid inheritance, small genome, easy transformation, high genetic diversity, and homology with legumes that have much larger genomes [25]. Abundant genome and genetic resources are available for *M. truncatula*, which is also valuable for the comparable studies of its close relative alfalfa.

Therefore, in the present study, we investigated the *CBL* and *CIPK* genes closely related to abiotic stress through comparative analysis between *M. sativa* and *M. truncatula*. We identified *CBL* and *CIPK* genes from both plant species, and analyzed their multiple sequence alignment, phylogenetic relationship, gene structure, protein motifs, and *cis*-acting elements. Furthermore, the expression profiles of *CBLs* and *CIPKs* of *M. sativa* and *M. truncatula* in response to various abiotic stresses were analyzed. Our comparative studies on *CBL* and *CIPK* genes from *M. sativa* and *M. truncatula* will provide a basis for future investigation on abiotic stress responses medicated by *CBLs* and *CIPKs*, and will facilitate the functional characterization of individual *CBLs* and *CIPKs* genes in responses to stresses and developmental signals.

## 2. Results

### 2.1. Identification of CBL-CIPK Genes in the M. sativa and M. truncatula Genome

Based on domain confirmation and homology search, a total of 10 *CBLs* and 26 *CIPKs* from *M. sativa*, and 13 *CBLs* and 32 *CIPKs* from *M. truncatula*, were identified, respectively. Their sequence were obtained from the *M. sativa* and *M. truncatula* genome database and further analyzed. Physico-chemical properties of *CBL* and *CIPK* genes, including TIGR locus, homologous gene, molecular weights, isoelectric points, possible subcellular localization were listed in Table 1. The encoded protein length of Ms/Mt *CBL* and Ms/Mt *CIPK* genes range from 173–336/191–257 aa and 207–519/237–518 aa. We also found the corresponding MW of Ms/Mt *CBL* and Ms/Mt *CIPK* ranged from 19.49–38.24/21.89–29.50 kDa and 25.00–58.00/23.36–57.89 kDa.

In addition, the isoelectric point (pI) of the Ms/Mt *CBL* and Ms/Mt *CIPK* ranges from 4.55–6.20/4.44–5.02 and 4.65–9.29/5.69–9.44, respectively (Table 1 and Table 2). The corresponding homologous *CBL* and *CIPK* genes of *M. sativa* and *M. truncatula* were identified in parallel by sequence alignment. The predicted subcellular locations suggested that most of the CBL and CIPK proteins from *M. sativa* and *M. truncatula* were located in the plasma membrane or extracellular (Table 1 and Table 2).

### 2.2. Multiple Sequence Alignment, Phylogenetic Analysis and Classification of CBL and CIPK Genes

Sequence alignments of the multiple amino acids between *MsCBLs* and *MtCBLs* indicated that the sequences of *MsCBLs* and *MtCBLs* are highly conserved (Appendix A): all the *CBLs* containing four EF hand motifs and one PFPF motif, which are similar to the *CBLs* from *Arabidopsis* [6]. Among them, *MsCBL2* had no EF hand 4 motif and PFPF motif, *MtCBL3* had no PFPF motif, and *MsCBL8* had no EF hand 2 and 3 motifs.

Similarly, the alignment results showed that all the CIPKs contain an N-terminal catalytic kinase domain and a C terminal regulatory domain, which are jointed by a variable domain (Appendix A). Correspondingly, the multiple amino acids of CIPK includes four domains: ATP binding site motif, activation loop motif, NAF motif and PPI motif [9]. Among them, *MsCIPK23*, *MsCIPK26*, and *MtCIPK11* do not have ATP binding site motif. *MsCIPK24* and *MtCIPK22* do not have the activation loop motif, and they are homologous genes. *MsCIPK10*, *11*, *12*, *13*, *14* do not have NAF motif, although they have been reported to be necessary to mediate interactions between CBL and CIPK proteins. In addition, *MsCIPK15*, *23*, and *MtCIPK10*, *11*, *12*, *13*, *14*, *22* do not contain PPI motif.

To investigate the evolutionary relationships of CBL and CIPK between *M. truncatula* and *M. sativa* and proteins from other plant species, a phylogenetic tree was constructed using the full amino acid sequences of CBL and CIPK family proteins from *M. sativa*, *M. truncatula*, and *A. thaliana*. It was shown that all of the CBLs could be classified into five distinct groups (I to V, Figure 1a), and all CIPKs into seven distinct groups (group A to G, Figure 1b) based on their sequence similarity, respectively. All of the *CBL* and *CIPK* genes from *M. sativa* and *M. truncatula* have their corresponding homologous genes.

Most *CBL* and *CIPK* genes in *M. sativa* and *M. truncatula* have homologous genes in *Arabidopsis*. The largest cluster for *CBL* was group I with 15 members, and the smallest cluster was group V with two members. However, there was no *CBL* member from *Arabidopsis* in group V, only a pair of homologous genes *MsCBL9* and *MtCBL8*. In addition, the largest group of *CIPK* are group A and G with both 18 members, and the smallest group is D, with only 6 *CIPK* members. In group B, 9 *CIPK* members of *M. sativa* and *M. truncatula* were closely related, but no *AtCIPK* members was found in this group.

### 2.3. Analyses of Conserved Motif and Gene Structure

Conserved motifs and intron/exon organization were analyzed in order to further investigate the structural features of CBLs and CIPKs of *M. sativa* and *M. truncatula* (Figure 2a). In the CBL proteins, motifs 1–3, 5, 7–8 corresponded to the conserved domain EF hand, while motif 4 corresponded to the conserved domain PFPF motif. Motif 1, 5, 8, 9 of CIPK proteins play key roles, which correspond to activation loop, ATP binding site, PPI motif, and NAF motif, respectively. The conserved NAF motif mediates CBL and CIPK physical interactions [9], indicating that the identified CIPKs may be functionally linked in the CBL and CIPK network. Moreover, the deletion of NAF motif gene was only found in group C, which may be related to classification and function. However, they all have motif 1 of the activation loop or motif 5 of the ATP binding site domain. Therefore, all the 8 CIPKs identified in this study showed conserved features of the CIPKs family, which is consistent with the motif analysis of CIPKs in *Arabidopsis* [2].

The gene structure of *CBLs* and *CIPKs* was analyzed, and it showed that each *CBL* has multiple introns. However, CIPK family members were clustered into an intron-rich clade (≥5 introns per gene) and an intron-poor clade (≤2 introns per gene), the intron-rich *CIPK* genes were clustered in subgroup F and G, while the intron-poor genes are presented in the other five subgroups (A–E). The structural differences in the *CIPKs* may allow *CIPK* genes to function differently because of the functional domains determine the function of genes [9].

### 2.4. Analysis of Collinearity and Chromosome Location of CBL and CIPK Genes

To further investigate the evolutionary mechanism of *CBL* and *CIPK* gene family, both tandem and segmental duplication events were analyzed. Overall, the distribution of *CBL* and *CIPKs* are not even in both *M. truncatula* and *M. sativa*. *MtCBLs* were distributed on six out of all eight chromosomes. There are four *MtCBL* genes in chromosome 2, three in chromosome 4 and 8, and one in chromosome 1, 3, 5; but none in chromosome 6 or 7 (Figure 3a). Unlike *M. truncatula*, *CBL* genes in *M. sativa* distributed only on 5 chromosomes, but not on chromosomes 1, 6, and 7 (Figure 3b). Additionally, four, two, two, one, and one *CBLs* genes were on chromosomes 8, 3, 5, 2, and 4 of *M. sativa*, respectively (Figure 3b).

For the *CIPK* genes, except chromosome 6 that did not have any gene for both *M. truncatula* and *M. sativa* (Figure 3b), all the other chromosomes have two to eight *CIPKs* for both *M. truncatula* and *M. sativa* (Figure 3b). In comparison, chromosome 3 of *M. truncatula*, and chromosome 8 of *M. sativa* had the most *CIPK* genes (Figure 3).

In addition, one *CBL* gene pair and one *CIPK* gene pair could be identified as segmental duplication events, two for *MtCBL*, four for *MtCIPK*, and one for *MsCIPK* were found to be tandem duplication in *M. truncatula* and *M. sativa* (Figure 3a,b). All above results inferred that both segmental duplication and tandem duplication events played an important driving force for the evolution of *CBL* genes, and the later played a predominant role. It is worth mentioning that although *M. sativa* and *M. truncatula* each had a pair of duplication genes, the number of tandem duplication genes in *M. truncatula* is far more than that of *M. sativa*.

Furthermore, three comparative syntenic maps of *M. sativa* and *M. truncatula* associated with the representative plant species *Arabidopsis* were constructed to illustrate the evolution relationship of *CBL* and *CIPK* genes family (Figure 3c). Notably, three orthologous pairs and 28 orthologous pairs were found between *M. truncatula* and *Arabidopsis*, and *M. truncatula* and *M. sativa*, respectively (Appendix A). There were two genes in *M. truncatula* which showed a collinear relationship with *Arabidopsis* and *M. sativa*, respectively. *MtCBL7* was collinear with *MsCBL5* and *MsCBL8*, and *MsCBL10* was collinear with *MtCBL2* and *MtCBL9*. Correspondingly, *MtCIPK21* and *MtCIPK23* were found to be associated with two collinear gene pairs in *Arabidopsis* and *M. sativa*, which may play a special role in the growth and development of *Medicago*.

Moreover, to better understand the evolutionary selection pressure during the formation of *CBL* and *CIPK* gene families, the Ka/Ks values of *CBL* and *CIPK* gene pairs were analyzed for both *M. sativa* and *M. truncatula* (Appendix A). Among all the *CBL* and *CIPK* gene pairs that are repeated in tandem and in segments, as well as the *M. sativa* and *M. truncatula* orthologous gene pairs, except for a homologous gene pairs (*MtCBL8*-*MsCBL9*), the Ka/Ks values of the remaining gene pairs are all less than 1. Taken together, these results indicated that the *CBL* and *CIPK* genes of *M. sativa* and *M. truncatula* may have undergone strong purification selection pressure during evolution.

### 2.5. Analyses of the Cis-Acting Element and Location of CBL and CIPK Genes

The *cis*-acting element is important for the binding of transcription factors, which control the expression of many target genes. Here, we focused on hormones and abiotic stress related *cis*-acting elements, including auxin responsive (AuxRE-core), gibberellin-responsive (GARE-motif, P-box, TATC-box), MeJA-responsive (TGACG-motif, CGTCA-motif), abscisic acid-responsive (ABRE), ethylene-responsive (ERE), salicylic acid responsiveness (TCA-element), defense and stress responsiveness (TC-rich repeats, W-box), wound responses (WUN motif), MYB binding site involved in drought-inducibility (MBS), low temperature-responsive (LTR), and anaerobic induction (ARE) (Figure 4 and Appendix A).

Our results showed that the promoter of *CBLs* and *CIPKs* genes contained various *cis*-acting elements with different number. In particular, many of them have stress-related *cis*-acting elements including ABRE, W-box, and MBS repeats (Figure 4). Among them, *MtCBL3*, *6*, *MsCBL3*, *9* have at least three ABRE repeat elements (Figure 4b,c). In addition, *MtCIPK5*, *MsCIPK6*, *MsCIPK10* contain seven, eight, and ten W-box repeat elements, respectively (Figure 4b,c) and *MtCIPK15*, *16*, *19*, *28*, *MsCIPK5*, *7*, contain at least three W-box repeat elements. Moreover, only *MtCIPK7* contained three MBS elements, and all the other genes had less than three MBS elements (Figure 4b,c). These genes are likely inducible by various stresses.

### 2.6. Analysis of the Expression Levels of CBL and CIPK Genes from Microarray Data

It is well known that CBL-CIPK complexes play important roles in the response of plants to external stimuli from environment [20,21,26]. Therefore, we retrieved genechip data of *M. truncatula* from the MtGEA web server (root under culture salinity, root under hydroponic salinity, root and shoot under drought treatment), and analyzed gene expression profiles in *M. truncatula*. One representative probe was selected for each gene, and the expression level of the representative probe is relatively close to the average value of multiple probes (Appendix A).

To further screen the stress responsive genes under different stress treatments, the average change folds of each gene under individual treatment (total change folds for all treatments/number of treatments) of the *MtCBL* and *MtCIPK* genes in each treatment under different stress treatments was calculated (Figure 5 and Appendix A). In addition, we also counted the number of treatments for each gene that were up-regulated by more than 2 times, which is consistent with the results of the above genes (Figure 5). Our expression analysis indicated that many *CBL* and *CIPK* genes are induced by these stresses (Figure 5).

Under drought treatment, 13 genes (*MtCIPK1*, *MtCBL6*, *MtCIPK2*, etc.) and 18 genes (*MtCIPK32*, *MtCBL9*, *MtCBL29*, etc.) were high induced in roots or in shoots, respectively (Figure 5a,b). In details, the expression levels of these genes were significantly increased under drought stress, but decreased after re-watering. It is worth noting that *MtCBL6* and *MtCBL9* were the only two *CBL* genes that were highly induced upon drought treatment, whether in roots or in stems (Figure 5a,b).

In addition, the gene expression levels were investigated in the roots that were treated with NaCl under culture and hydroponic condition, and it was found that 16 genes (*MtCIPK2*, *MtCIPK8*, *MtCIPK21*, etc., Figure 5c) and 18 genes (*MtCIPK22*, *MtCIPK26*, *MtCIPK18*, etc., Figure 5d) were highly induced under these two conditions, respectively. Notably, all of them were *CIPK* genes except one *CBL* (*MtCBL12*) gene that were highly induced under the culture NaCl treatment condition.

Moreover, Venn diagram, showing the differential expression level of all genes under four different treatments (Figure 5a–d), demonstrated that 17 genes were highly expressed under both drought and NaCl treatment (in red in Figure 5e). In particular, *MtCIPK1*, *3*, *17*, *18*, *21* showed relatively higher expression level than the other genes under four different treatments (Figure 5f).

Due to the close relationship between gene expression level and gene function, we also investigated the expression profiles of *CBLs* and *CIPKs* in seven tissues of *M. truncatula* (root, stem, leaf, flower, pod, seed, and bud) with available microarray data (Figure 5g). Remarkably, 12 genes (*MtCIPK1*, *2*, *7*, *8*, *23*, *26, 27, 28, 29*, *32*, and *MtCBL7*, *10*) showed relatively high expression level in various tissues. On the contrary, 6 genes (*MtCBL2*, *6*, *8*, and *MtCIPK3*, *17*, *22*) were expressed at relatively low level in different tissues (Figure 5g). Besides, the expression levels of the remaining 12 genes (*MtCIPK1*, *9*, *12*, *13* and *MtCIPK18*–*21*, *24*, *25*, *30*, and *31*) showed different pattern in various tissues. Altogether, the expression profiles of *CBLs* and *CIPKs* genes varied greatly, suggesting that they may be functional in different tissues.

### 2.7. Validation of the Expression Profile of Stress-Responsive CBL and CIPK Genes by qPCR Analysis

In order to verify the expression profiles of *MtCBL* and *MtCIPK* genes from microarray data, qPCR verification were carried out with ten selected genes. Among them, three *MtCIPK* genes (*MtCIPK2*, *26*, *28*) and two *MtCBL* genes (*MtCBL7*, *10*) were selected among the highly expressed genes as representatives (Figure 5g). In addition to these five genes, the other five genes (*MtCIPK1*, *3*, *17*, *18*, *21*) were also selected for qPCR verification because they were highly inducible under stresses (Figure 5f). In addition, their homology genes from *M. sativa*, as shown in Table 1 and Table 2, were also selected as gene pairs for qPCR verification for comparison purpose.

The expression of these twenty genes were analyzed under two stresses (NaCl and PEG) and ABA treatments (Figure 6). Overall, most genes were strongly induced by multiple treatments. In more details, under PEG stress, all of these genes except *MsCIPK25*, were significantly up-regulated to different levels, although their patterns are not the same for *M. truncatula* or *M. sativa* at different time points (Figure 6, left). Under NaCl treatment, many of the genes were up-regulation at different level and time points, but *MtCIPK1*–*3*, *8* and *MtCBL10* particularly were significantly up-regulated at 72 h (Figure 6, middle). Upon ABA treatment, the fold changes of all genes were not more than 15 folds, with many of them around 6 folds (Figure 6, right). However, the expression levels of *MsCBL5*, *6*, *MsCIPK2*, *4*, *11*, *25* did not show any significant changes, indicating they were not induced by ABA treatment (Figure 6, right).

## 3. Discussion

The signaling system composed of CBLs and CIPKs as a calcium sensors is a key regulatory node in stress signaling pathways in plant. Significantly, regulation of membrane transport processes seems to be an emerging theme in the function of CBL-CIPK signaling machinery [19]. Extensive evidence confirmed the idea that CBL-CIPK proteins network regulates abiotic stress in many species, but little is known in *Medicago*. In the current study, a search for *CBL* and *CIPK* genes brought about the identification of 10 *MsCBL*, 13 *MtCBL*, 26 *MsCBL*, and 32 *MtCBL* genes in *M. sativa* and *M. truncatula*, respectively.

Multiple sequence alignments showed that most of the *Medicago* CBL proteins contained four EF hand motifs, which are necessary for CBL proteins to bind Ca^2+^, and the linker between each EF hand motif is absolutely conserved among CBL proteins [27]. Among them, *MsCBL8* had no EF hand 2 and 3 motifs, and the phylogenetic analysis of *MsCBL8* did indicate that it has high homology with *AtCBL1* and *AtCBL9* protein (Appendix A), suggesting that it might have specific functions and needs further study. Most CIPK proteins have been demonstrated to contain two domains: An N-terminal kinase catalytic domain (NAF motif) and a C-terminal regulatory domain (PPI motif) (Appendix A). However, *MsCIPK10*, *11*, *12*, *13*, *14* do not have NAF motif or PPI motif, and the phylogenetic analysis of them clustered together. Therefore, they may perform special functions because they contain the ATP binding site domain and activation loop domain, belonging to the CIPK family. Phylogenetic analysis showed that CBLs and CIPKs had an independent *Medicago* cluster without *Arabidopsis* members (cluster V for CBL, cluster B for CIPK). Other clustering patterns were highly consistent with *Arabidopsis* [2]. Therefore, these members play important roles in the evolution of CBLs and CIPKs families, respectively.

Conserved motif identification shows that most *CIPKs* identified have a NAF domain in the C-terminal region (Figure 2b). It can be seen that they are close phylogenetically, and the types and positions of their motifs are more similar. Some motifs only exist in part of the *CBL* or *CIPK* amino acid sequence. For example, motif 5 belongs to the EF hands domain of CBL protein that only appears in one group I (Figure 2a), and these findings suggested that these motifs may be related to the functional diversity of CBL proteins. Gene structures like intron/exon organizations often reflect the evolution within some gene families [28,29,30]. Gene structure analysis showed that *CBLs* have more introns, among them, *MsCBL3* have 10 introns, while *MsCBL8, 9, 10* have 4 introns (Figure 2a). The phylogenetic relationship analysis showed that they belong to different groups, and the functions of them might be different, as shown in Figure 2b. Interestingly, CIPKs can be divided into two types: Poor introns (A–E) and rich introns (F,G). Similar patterns of intron-rich/poor CIPK family members have also been observed in model plants, *Arabidopsis*, rice, poplar, and soybean [12,31,32]. Significantly, there are abundant introns in CIPK proteins of green algae [33], moss, and ferns, and the intron deletion of CIPK first occurs in seed plants, which may be due to the species evolution caused by intron deletion in intron-rich members [2].

Duplication and divergence play an important role in expansion and evolution of gene families [34,35]. We found one (3.8%) segmental duplication and tandem duplication among 26 *MsCIPKs*, and one (10%) segmental duplication in ten *MtCBLs*, three (30%) tandem duplication in ten *MtCBLs* and four (12.5%) tandem duplication among thirty-two *MtCIPKs* (Figure 3). It shows that gene duplication is the main evolutionary force for the expansion of *CBL* and *CIPK* gene families in *Medicago*. Based on gene expression, combined with phylogenetic analysis and collinearity analysis, it is valuable to understand the functions of *CBL* and *CIPK* genes in special physiological processes. For example, *MtCIPK31* has the highest expression level in seeds and flowers, but is relatively low in other tissues, which is consistent with its tandem duplication *MtCIPK30* (Figure 5g), it is worth noting that there is a collinearity between *MsCIPK21* and *MtCIPK30*. Interesting, their closest ortholog gene was *AtCIPK9* in *Arabidopsis*, which is expressed predominantly in root, where it regulates K^+^-homeostasis under low-K^+^ stress in *Arabidopsis* [36,37]. This result indicated that *MtCIPK30* and *MtCIPK31* may share similar function in regulate K^+^-homeostasis under abiotic stress in *Medicago*. Another orthologous pair *MsCIPK22* and *MsCIPK23* were homologies of *OsCIPK9* that were induced by multiple stresses, including cold, drought, PEG, and ABA treatments [38], suggesting the potential function of *MsCIPK22* and *MsCIPK23* in participating abiotic stress.

High-salinity or drought soil is the most serious abiotic stress [39]. It is urgent to improve the salinity and drought tolerance of alfalfa to increase yield. The transcript data from genechip under NaCl and drought treatments suggested that the expressions of several *CBL* and *CIPK* genes were highly induced or drastically changed (Figure 5). Simultaneously, the expression pattern of all ten genes were verified to be up-regulated under several treatments in *M. truncatula* by qPCR analyses (Figure 6) and suggested that these ten genes are involved in plant abiotic stress and hormone induction. Correspondingly, three gene pairs (*MtCIPK2*-*MsCIPK3*, *MtCIPK17*-*MsCIPK11,* and *MtCIPK18*-*MsCIPK12*) were significantly up-regulated under PEG, NaCl, and ABA treatments, and their expression patterns were the same in *M. sativa* and *M. truncatula*. By identifying the *cis*-acting elements bound by specific transcription factors, it is possible to reveal the transcriptional regulatory mechanism and gene expression patterns of the plant environmental adaptation. Unsurprisingly, all these gene pairs contain at least two of abscisic acid-responsive (ABRE), defense and stress responsiveness (TC-rich repeats, W-box), and MYB binding site involved in drought inducible (MBS) (except for the *MtCIPK18* gene). Notably, *MsCIPK3* gene contains more defense and stress responsiveness elements. Consequently, these genes may play key roles in enhancing the stress resistance of *M. sativa* and *M. truncatula*.

Previous studies have confirmed that CBL, CIPK, or CBL-CIPK complex function in regulating the complex process of plant growth and responding to environmental stress [18,40]. *AtCIPK7* regulates cold stress responses via its interaction with *AtCBL1* [41]. However, the homologous genes *MtCIPK2* and *MsCIPK3* exhibited high expression under several stresses, indicating that these two genes are involved in other pathways except cold stress. *AtCIPK25* participates in the development of root meristem through auxin and cytokinin signaling [42]. The closely related genes of *MtCIPK18* and *MsCIPK12* may also play a similar role in resisting environmental adversity. *AtCBL1* was reported to interact with *AtCIPK1*, involved in ABA-dependent stress responses [43]. Moreover, *AtCBL1*/*9* can interact with *AtCIPK23* to regulate both K^+^ and NO^3−^ uptake in plants [44]. The ortholog gene *MtCBL7* (*AtCBL9*) was up-regulated by PEG, NaCl, and ABA stress in *M. truncatula* (Figure 6). Interesting, *AtCBL9*–*AtCIPK1* complex are also involved in ABA independent stress responses [43]. *OsCIPK1* from rice was speculated to connect ABA and cold signaling pathway [40]. The orthologous gene *MtCIPK21* (*AtCIPK1*) was up-regulated by almost all kinds of stresses in *M. truncatula*, and ABA-responsive element ABRE was found in the promoter region of *MtCIPK21* (Figure 4). These results suggested the potential roles of *CBL* and *CIPK* genes in *M. sativa* and *M. truncatula* under abiotic stresses-resistance and hormone-resistance. Meanwhile, these homologous genes of *M. sativa* and *M. truncatula* did not all exhibit the same expression pattern under different stress treatments. It is possible that these genes have changed to varying degrees during the species evolution.

## 4. Materials and Methods

### 4.1. Identification of CBL-CIPK Genes in the Medicago Genome

The genome sequences and deduced protein sequences of *CBLs* and *CIPKs* were downloaded from the *M. sativa* and *M. truncatula* genome website (http://www.medicagogenome.org/, accessed on 15 November 2020) [45]. The first step in determining the *CBL* (EF-hand calcium-binding domain (PS50222)) and *CIPK* (NAF domain (PF03822)) family is to download Hidden Markov Model (HMM) profiles from Pfam protein family database (https://pfam.xfam.org/, accessed on 16 November 2020). Secondly, the *CBL* and *CIPK* gene sequences from Arabidopsis were downloaded from TAIR (https://www.arabidopsis.org/, accessed on 16 November 2020). The *CBLs* and *CIPKs* sequences from *Arabidopsis* were used as a query (*p* < e^−5^) to search the protein sequence of *M. sativa* and *M. truncatula*. The identified sequences were further verified, and redundant sequences were removed from the list. Further, output putative CBL and CIPK protein sequences were submitted to InterProScan (https://www.ebi.ac.uk/interpro/search/sequence-search, accessed on 20 November 2020), CDD (https://www.ncbi.nlm.nih.gov/Structure/bwrpsb/bwrpsb.cgi, accessed on 20 November 2020), Pfam (https://pfam.xfam.org/, accessed on 22 November 2020), and SMART (http://smart.embl-heidelberg.de/, accessed on 22 November 2020) to confirm the conserved CBL and CIPK domain. Finally, all *CBL* and *CIPK* candidate genes were obtained and assigned based on their locations on chromosome, including 10 *MsCBL* genes, 13 *MtCBL* genes, 26 *MsCIPK* genes, and 32 *MtCIPK* genes. Correspondingly, ExPASy (https://web.expasy.org/compute_pi/, accessed on 24 November 2020) was used to determine the isoelectric point (pI) and molecular weight (MW) of CBL and CIPK proteins. Finally, subcellular localization of *CBL* and *CIPK* genes were predicted by using the Softberry Home Page (http://linux1.softberry.com/berry.phtm, accessed on 24 November 2020).

### 4.2. Sequence Analyses and Structural Characterization of Medicago CBL and CIPK Genes

Sequence alignment analysis of *CBL* and *CIPK* domain sequences were carried out by using jalview v2.10.5 (Solvusoft Corporation, Las Vegas, NV, USA). Conserved motifs in CBL and CIPK protein sequences were identified by the MEME program (MEME-Suite version 5.1.0, http://meme-suite.org/, accessed on 2 December 2020) with default settings, except the motif number of *CBL* was set as 10 and 20 for *CIPK*, and the width of minimum and maximum motif were set as 10 and 200, respectively. The visualization of exon–intron positions and conserved motifs was executed through Amazing Optional Gene Viewer software (TBtools v1.068, South China Agricultural University, Guangzhou, Guangdong, China) [46].

### 4.3. Phylogenetic Analysis and Classification of the CBL and CIPK Genes

The phylogenetic relationship of CBL-CIPK proteins among *M. sativa*, *M. truncatula*, and *Arabidopsis* were analyzed with identified CBL-CIPK amino acid sequences from *M. sativa*, *M. truncatula*, and *Arabidopsis*. The phylogenetic trees were constructed by using MEGA-X (Mega Limited, Auckland, New Zealand) [47] using the Neighbor-Joining (NJ) method with default parameters with bootstrap value of 1000. Finally, EvolView (https://evolgenius.info/evolview-v2/, accessed on 5 December 2020) was used to modify the evolutionary tree.

### 4.4. Analysis of Collinearity and Chromosome Location

The chromosomal localization of all *CBLs* and *CIPKs* of *M. sativa* and *M. truncatula* were obtained from the corresponding website. Multiple collinear Scan toolkit (TBtools v1.068 (mcscanx), South China Agricultural University, Guangzhou, Guangdong, China) was used to analyze the gene duplication events with default parameters [48]. All the *CBL* and *CIPK* genes were similarly mapped to eight *M. sativa* and *M. truncatula* chromosomes, respectively, followed by the analysis on their intraspecific synteny relationship in *M. sativa* and *M. truncatula* using Amazing Gene Location software (TBtools), respectively [46]. In order to exhibit the interspecific synteny relationship among *M. sativa*, *M. truncatula*, and the representative model plant species (*Arabidopsis*), the syntenic maps were constructed using the Dual Systeny Plotter software (TBtools) [46]. The simple Ka/Ks calculator software is used to calculate non synchronous (Ka) and synchronous (Ks) values of *CBL* and *CIPK* gene pairs (TBtools) [46].

### 4.5. Analyses of the Cis-Acting Elements and Location of CBL and CIPK Genes in Medicago

The *cis*-acting elements in the 2000 bp upstream sequences of the coding region of *CBL* and *CIPK* genes were analyzed using PlantCARE (http://bioinformatics.psb.ugent.be/webtools/plantcare/html/, accessed on 10 December 2020) [49]. TBtools was used to visualize the *cis*-acting elements *CBL* and *CIPK* genes of *Medicago*. 

### 4.6. Analysis of the Expression Levels of CBL and CIPK Genes from Genechip Data

For the analysis of the expression level of *CBL* and *CIPK* genes from microarray data, we download all the genechip data from *M. truncatula* Gene Expression Atlas (https://Mtgea.noble.org/v3/, accessed on 12 December 2020)), which has been developed as a compendium or “atlas” of gene expression profiles for the *M. truncatula* genes. The selected genechip data covered the roots and shoots, from plants subjected to drought stress, salt stress, and ABA stresses, and specific cell and tissue types. Amazing HeatMap software was used to generate the heatmap (TBtools) [46].

### 4.7. Plant Materials and Treatments

The *M. sativa* (Zhongmu No.1, Institute of Animal Science of CAAS, Beijing, China) and *M. truncatula* (cv. Jemalong A17, Institute of Animal Science of CAAS, Beijing, China) plants used in this study was stored at the Institute of Animal Science of CAAS. The roots, stems, leaves, flowers, pods (20-day old pods), and seeds (20-day old seeds) of mature *M. sativa* and *M. truncatula* plants, were collected separately for RNA extraction and used for qPCR analysis. Meanwhile, the expression patterns of *CBL* and *CIPK* genes were also investigated under different stress and hormone treatments (PEG, NaCl and ABA). The material and method used for different treatments were the same for *M. sativa* as for *M. truncatula* as previously reported by Yang et al. [50].

### 4.8. Analysis of the Gene Expression by qRT-PCR

Total RNA extraction, first strand cDNA synthesis, and qPCR procedures and methods were the same as previously reported by Yang et al. [50]. Each reaction was performed in biological triplicates and the data from qPCR was analyzed using 2^−ΔΔCT^ method. The results were analyzed by means ± standard deviation (SD). The primer sequences used in this study were shown in details in Appendix A.

## 5. Conclusions

The CBL-CIPK network is well-recognized as one of the resistance mechanisms. This study analyzed the CBL and CIPK genes on a genome-wide scale in *M. sativa* and *M. truncatula*. A total of 10 *MsCBLs*, 13 *MtCBLs*, 26 *MsCIPKs*, and 32 *MtCIPKs* were identified, respectively, in *M. sativa* and *M. truncatula*. These genes show high similarity in amino acid sequence, motif compositions, and conservative gene structure. In addition, phylogenetic analysis and collinearity analysis on *CBL* and *CIPK* in different species revealed their evolutionary patterns and predicted their functions in complex environments. Moreover, the expression profile of *CBL* and *CIPK* genes in different tissues and treatments of *Medicago* were analyzed and were verified by qPCR. It was found that most of the genes were highly expressed, especially *MtCIPK2*-*MsCIPK3*, *MtCIPK17*-*MsCIPK11,* and *MTCIPK18*-*MsCIPK12*. These gene pairs showed the same expression pattern in *M. sativa* and *M. truncatula*, and they may play an important role in response of plant stresses. This study compares the CBL-CIPK family of *M. sativa* and *M. truncatula* to provide new ideas for understanding the evolutionary relationship between the same genus and related crops, which also provides a comprehensive information basis for better characterizing the role of the CBL-CIPK family in plant stress resistance.

## Figures and Tables

**Figure 1 ijms-22-04634-f001:**
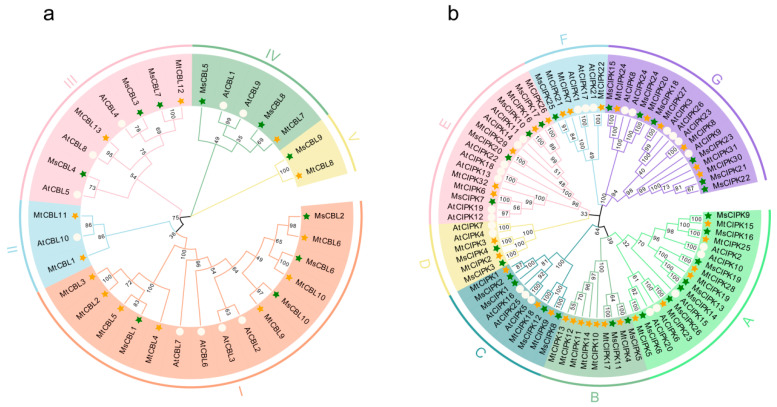
The phylogenetic analysis of CBL (**a**) and CIPK (**b**) proteins families across *Medicago* and *Arabidopsis*. Full length protein sequences of CBLs and CIPKs were constructed using MEGA-X based on the Neighbor-Joining (NJ) method; bootstrap was 1000 replicates. Subfamilies are highlighted with different colors. The green solid pentagrams, orange solid pentagrams, and hollow circles represent CBL and CIPK proteins from *M. sativa* (Ms), *M. truncatula* (Mt), and *A. thaliana* (At), respectively.

**Figure 2 ijms-22-04634-f002:**
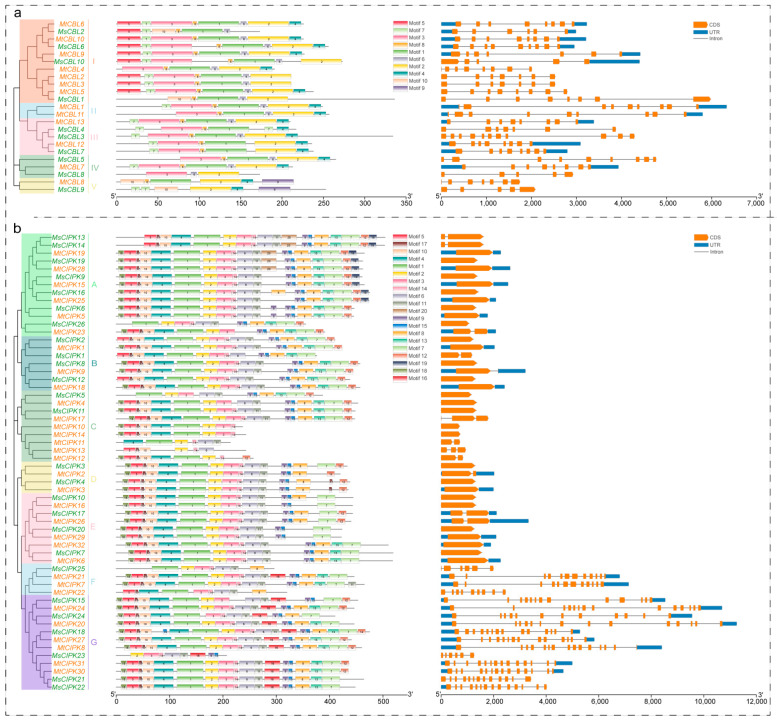
Analyses on phylogenetic relationships, motifs, and gene structure of *CBL* and *CIPK* genes from *M. sativa* and *M. truncatula*. The groups and its color of the phylogenetic tree are consistent with those in Figure 1. The motifs were indicated in different colored boxes with different numbers and the sequence information for each motif is provided in Appendix A. Exon–intron structure and conserved domain of *CBL* and *CIPK* genes. Blue boxes indicate 5′- and 3′- untranslated regions; orange boxes indicate exons; black lines indicate introns. (**a**,**b**) Phylogenetic relationships, motifs, and gene structure of *CBL* genes (**a**) and *CIPK* genes (**b**).

**Figure 3 ijms-22-04634-f003:**
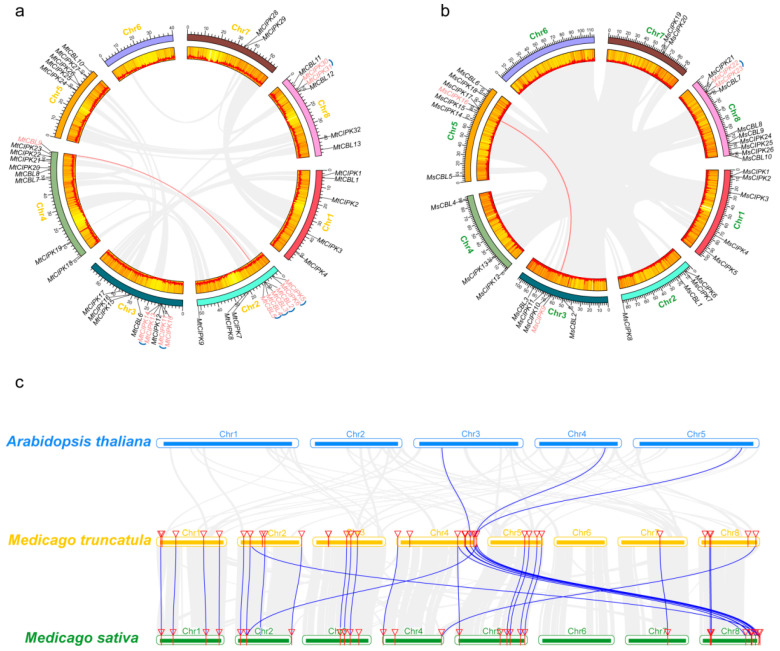
Chromosome distributions of *CBLs* and *CIPKs* in *M. truncatula* and *M. sativa*. The chromosomal location and interchromosomal relationships of *CBL* and *CIPK* genes. The tandem duplicated genes are marked by blue arc trajectory and segmentally duplicated genes are connected by red curves. The chromosomal location and interchromosomal relationships of *M. truncatula* (**a**) and *M. sativa* (**b**). (**c**) Synteny analysis of *CBL* and *CIPK* genes between *M. sativa*, *M. truncatula* and representative plant species (*Arabidopsis*). Gray lines in the background indicate the collinear blocks within *M. truncatula*, and *M. sativa*/*Arabidopsis*, and the blue lines highlight the syntenic *CBL* and *CIPK* gene pairs.

**Figure 4 ijms-22-04634-f004:**
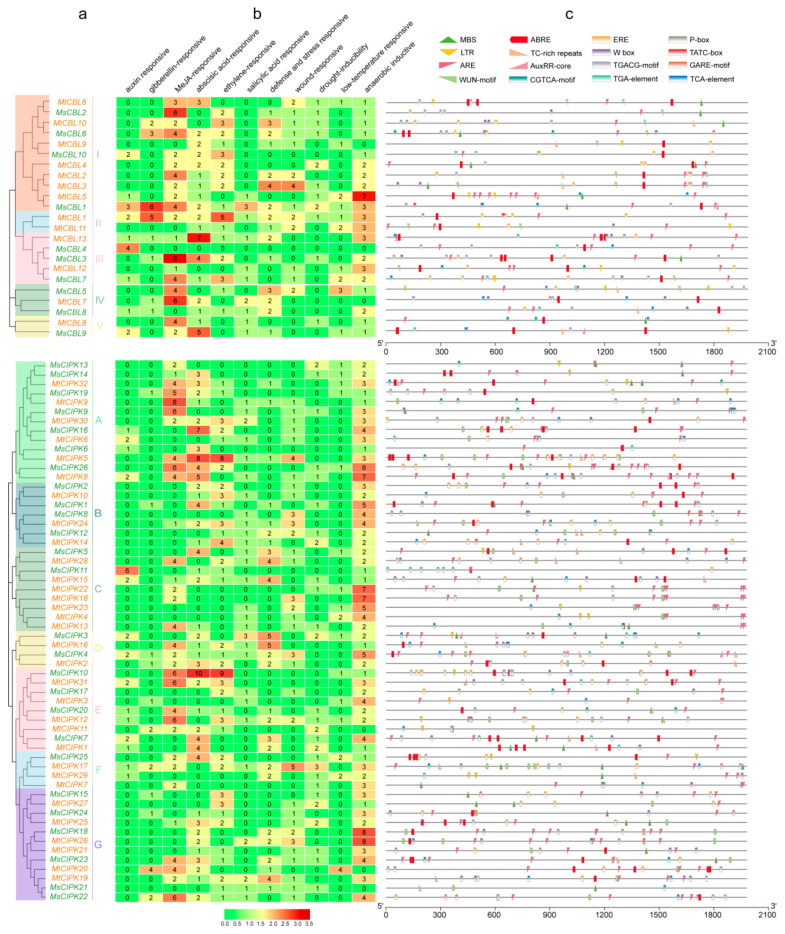
Putative cis-elements and transcription factor binding sites in the promoter regions of *CBL* and *CIPK* genes from *M. sativa* and *M. truncatula*. (**a**) The groups and color are indicated as in Figure 1. (**b**) The colors and numbers of the grid indicated the numbers of different *cis*-acting elements in these *CBL* and *CIPK* genes. (**c**) The colored block represented different types of *cis*-acting elements and their locations in each *CBL* and *CIPK* gene.

**Figure 5 ijms-22-04634-f005:**
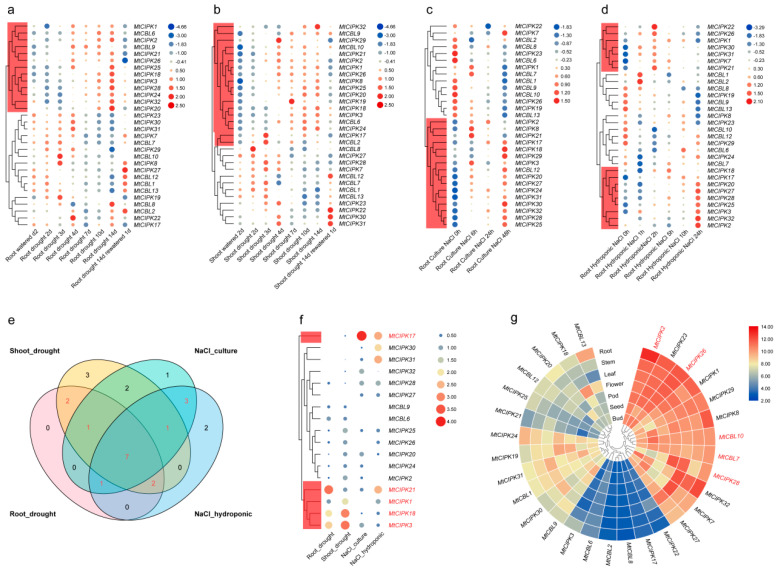
*MtCBL* and *MtCIPK* gene expression profiles under different treatments, different times, and different tissues retrieved from microarray data. Each column represents one treatment, each row represents one gene, and each member is normalized in the same column. The relative expression levels are log2-transformed and visualized for heatmap. The color of circles from blue to red shows the expression level from negative to positive values after normalization. The red box indicates genes with increased expression. (**a**) Gene expression of root under different drought treatment time. (**b**) Gene expression of shoot under different drought treatment time. (**c**) Gene expression of root under different NaCl-culture time. (**d**) Gene expression of root under different NaCl-hydroponics time. (**e**) Venn diagram of gene expression levels after four different treatments. (**f**) Gene expression levels of common differentially expressed genes after different treatments in the Figure 5e. (**g**) Expression profiles of genes in 7 different tissues retrieved from microarray data.

**Figure 6 ijms-22-04634-f006:**
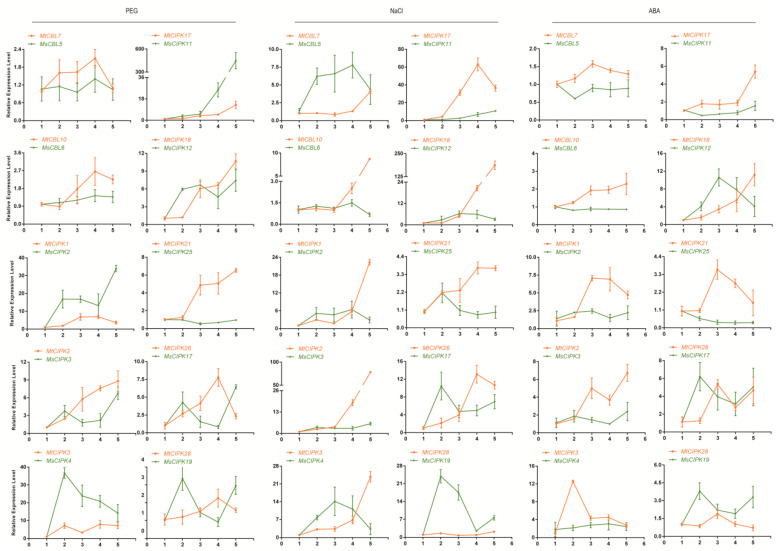
Quantification of gene expression levels of selected *CBL* and *CIPK* genes from *M. sativa* and *M. truncatula* under PEG, NaCl, and ABA stress using qPCR. Data are average of three independent biological samples ±SE and vertical bars indicate standard deviation. The value 1–5 of *X* axis for M. sativa represent treatment period of 0, 4, 6, 8, and 24 h, respectively. The value 1–5 of *X*-axis for *M. truncatula* represent treatment period of 0, 2, 24, 48, 72 h, respectively.

**Table 1 ijms-22-04634-t001:** Properties of the predicted CBL proteins in *M. sativa* and *M. truncatula.*

Gene Name	TIGR Locus	Homologous Gene	Chromosome Location	pI	MW	Protein Length	Subcellular Localization
*MsCBL1*	MsG0280007543.01.T01	*MtCBL5*	Chr2: 16946249–16952230	6.2	38,240.38	336	Extracellular
*MsCBL2*	MsG0380014586.01.T01	*MtCBL6*	Chr3: 56818145–56821980	4.9	19,486.31	173	Plasma membrane
*MsCBL3*	MsG0380016524.01.T01	*MtCBL7*	Chr3: 84678194–84682482	4.9	38,012.36	334	Plasma membrane
*MsCBL4*	MsG0480023701.01.T01	*MtCBL9*	Chr4: 88741570–88745444	4.55	24,633.18	217	Plasma membrane
*MsCBL5*	MsG0580024593.01.T01	*MtCBL7*	Chr5: 6687437–6692205	4.77	30,599.99	265	Plasma membrane
*MsCBL6*	MsG0580030002.01.T01	*MtCBL10*	Chr5: 104956837–104960723	5.02	29,541.79	256	Plasma membrane
*MsCBL7*	MsG0880043002.01.T01	*MtCBL12*	Chr8: 17451660–17454557	4.82	27,195.04	238	Plasma membrane
*MsCBL8*	MsG0880046091.01.T01	*MtCBL7*	Chr8: 68926482–68929407	4.99	20,024.43	173	Plasma membrane
*MsCBL9*	MsG0880046710.01.T01	*MtCBL8*	Chr8: 77208147–77210234	4.79	26,744.53	253	Extracellular
*MsCBL10*	MsG0880047760.01.T01	*MtCBL9*	Chr8: 90474978–90479379	5.64	31,753.31	273	Plasma membrane
*MtCBL1*	MtrunA17_Chr1g0151621	*MsCBL3*	Chr1: 4424275–4430607	4.68	28,348.38	249	Plasma membrane
*MtCBL2*	MtrunA17_Chr2g0290751	*MsCBL6*	Chr2: 10098104–10100632	4.8	24,076.53	211	Plasma membrane
*MtCBL3*	MtrunA17_Chr2g0290761	*MsCBL6*	Chr2: 10120014–10122541	5.02	24,212.82	211	Plasma membrane
*MtCBL4*	MtrunA17_Chr2g0290771	*MsCBL6*	Chr2: 10131045–10133055	4.44	21,889.77	191	Plasma membrane
*MtCBL5*	MtrunA17_Chr2g0290781	*MsCBL6*	Chr2: 10140913–10143707	4.76	27,082.64	238	Plasma membrane
*MtCBL6*	MtrunA17_Chr3g0102821	*MsCBL6*	Chr3: 27325210–27329496	4.7	25,954.5	226	Plasma membrane
*MtCBL7*	MtrunA17_Chr4g0055421	*MsCBL5*	Chr4: 49038532–49043130	4.74	24,483.05	213	Plasma membrane
*MtCBL8*	MtrunA17_Chr4g0063121	*MsCBL9*	Chr4: 54777738–54779483	4.54	23,911.69	214	Plasma membrane
*MtCBL9*	MtrunA17_Chr4g0076551	*MsCBL6*	Chr4: 64369731–64374408	4.75	25,972.49	227	Plasma membrane
*MtCBL10*	MtrunA17_Chr5g0446931	*MsCBL6*	Chr5: 43351610–43355760	4.71	26,064.69	226	Plasma membrane
*MtCBL11*	MtrunA17_Chr8g0341941	*MsCBL3*	Chr8: 5383017–5389089	4.78	29,496.87	257	Plasma membrane
*MtCBL12*	MtrunA17_Chr8g0346911	*MsCBL7*	Chr8: 9926428–9929518	4.99	27,094.98	236	Plasma membrane
*MtCBL13*	MtrunA17_Chr8g0388331	*MsCBL3*	Chr8: 46203348–46207118	4.97	24,028.73	210	Plasma membrane

**Table 2 ijms-22-04634-t002:** Properties of the predicted CIPK proteins in *M. sativa* and *M. truncatula.*

Gene Name	TIGR Locus	Homologous Gene	Chromosome Location	pI	MW	Protein Length	Subcellular Localization
*MsCIPK1*	MsG0180000513.01.T01	*MtCIPK1*	Chr1: 7219758–7220941	8.93	42,516.31	376	Plasma membrane
*MsCIPK2*	MsG0180000521.01.T01	*MtCIPK1*	Chr1: 7356600–7357829	8.99	46,413.89	410	Plasma membrane
*MsCIPK3*	MsG0180001670.01.T01	*MtCIPK2*	Chr1: 24938893–24940191	8.85	48,636.26	433	Plasma membrane
*MsCIPK4*	MsG0180004166.01.T01	*MtCIPK3*	Chr1: 74311950–74313263	8.86	48,883.24	438	Plasma membrane
*MsCIPK5*	MsG0180005611.01.T01	*MtCIPK4*	Chr1: 93961932–93963092	8.09	44,450.35	387	Extracellular
*MsCIPK6*	MsG0280006904.01.T01	*MtCIPK5*	Chr2: 7792269–7793606	8.87	51,100.97	446	Plasma membrane
*MsCIPK7*	MsG0280006906.01.T01	*MtCIPK6*	Chr2: 7819514–7821070	6.56	57,894.45	519	Plasma membrane
*MsCIPK8*	MsG0280011471.01.T01	*MtCIPK9*	Chr2: 84572890–84574260	9.15	51,553.44	457	Plasma membrane
*MsCIPK9*	MsG0380014945.01.T01	*MtCIPK15*	Chr3: 62899825–62901207	8.58	52,512.51	461	Plasma membrane
*MsCIPK10*	MsG0380014950.01.T01	*MtCIPK16*	Chr3: 62944869–62946200	8.45	50,073.61	444	Plasma membrane
*MsCIPK11*	MsG0380015546.01.T01	*MtCIPK17*	Chr3: 71653648–71654991	8.94	51,591.75	448	Plasma membrane
*MsCIPK12*	MsG0480018187.01.T01	*MtCIPK18*	Chr4: 1337477–1338790	8.58	50,031.2	438	Plasma membrane
*MsCIPK13*	MsG0480019352.01.T01	*MtCIPK19*	Chr4: 18884311–18885931	8.55	56,612.39	504	Plasma membrane
*MsCIPK14*	MsG0580027748.01.T01	*MtCIPK19*	Chr5: 67945209–67946826	8.36	56,525.14	503	Plasma membrane
*MsCIPK15*	MsG0580028307.01.T01	*MtCIPK24*	Chr5: 78145226–78153759	6.37	51,712.85	453	Plasma membrane
*MsCIPK16*	MsG0580028588.01.T01	*MtCIPK25*	Chr5: 82921999–82923423	9.05	53,886.01	475	Plasma membrane
*MsCIPK17*	MsG0580028590.01.T01	*MtCIPK26*	Chr5: 82973828–82975942	7.14	49,482.95	443	Plasma membrane
*MsCIPK18*	MsG0580029551.01.T01	*MtCIPK27*	Chr5: 98438716–98444881	6.45	53,945	475	Plasma membrane
*MsCIPK19*	MsG0780039388.01.T01	*MtCIPK28*	Chr7: 63273062–63274447	8.02	52,396.38	462	Plasma membrane
*MsCIPK20*	MsG0780039391.01.T01	*MtCIPK29*	Chr7: 63319880–63321148	8.95	47,583.98	423	Plasma membrane
*MsCIPK21*	MsG0880042910.01.T01	*MtCIPK30*	Chr8: 15785566–15788993	9.03	52,865.88	464	Plasma membrane
*MsCIPK22*	MsG0880042931.01.T01	*MtCIPK30*	Chr8: 16193214–16197233	8.99	50,776.14	448	Plasma membrane
*MsCIPK23*	MsG0880042932.01.T01	*MtCIPK31*	Chr8: 16202211–16203469	4.65	23,360.42	207	Plasma membrane
*MsCIPK24*	MsG0880046785.01.T01	*MtCIPK20*	Chr8: 78103239–78112793	9.29	46,753.34	410	Plasma membrane
*MsCIPK25*	MsG0880047252.01.T01	*MtCIPK21*	Chr8: 83803682–83805677	9.18	33,282.91	296	Plasma membrane
*MsCIPK26*	MsG0880047603.01.T01	*MtCIPK23*	Chr8: 88629631–88630695	8.59	40,180.45	355	Plasma membrane
*MtCIPK1*	MtrunA17_Chr1g0150321	*MsCIPK2*	Chr1: 3363660–3365695	6.51	57,997.54	518	Plasma membrane
*MtCIPK2*	MtrunA17_Chr1g0166101	*MsCIPK3*	Chr1: 15531311–15533332	8.71	48,428.7	435	Plasma membrane
*MtCIPK3*	MtrunA17_Chr1g0188031	*MsCIPK4*	Chr1: 37868206–37870202	7.6	49,372.84	440	Plasma membrane
*MtCIPK4*	MtrunA17_Chr1g0205421	*MsCIPK5*	Chr1: 50641418–50642779	9.44	28,820.94	243	Plasma membrane
*MtCIPK5*	MtrunA17_Chr2g0284011	*MsCIPK6*	Chr2: 4993431–4995206	8.77	50,920.75	445	Plasma membrane
*MtCIPK6*	MtrunA17_Chr2g0284031	*MsCIPK7*	Chr2: 5004740–5007006	8.92	53,811.89	474	Plasma membrane
*MtCIPK7*	MtrunA17_Chr2g0302431	*MsCIPK22*	Chr2: 19969671–19976812	7.56	36,077.73	320	Plasma membrane
*MtCIPK8*	MtrunA17_Chr2g0304111	*MsCIPK18*	Chr2: 21918963–21927365	8.71	44,042.03	391	Plasma membrane
*MtCIPK9*	MtrunA17_Chr2g0333821	*MsCIPK8*	Chr2: 51523898–51527942	7.99	52,575.69	462	Plasma membrane
*MtCIPK10*	MtrunA17_Chr3g0091281	*MsCIPK11*	Chr3: 12548682–12549395	8.9	48,125.84	423	Plasma membrane
*MtCIPK11*	MtrunA17_Chr3g0091291	*MsCIPK11*	Chr3: 12550018–12550736	8.44	57,285.85	510	Plasma membrane
*MtCIPK12*	MtrunA17_Chr3g0091311	*MsCIPK11*	Chr3: 12582134–12582970	8.95	47,560	422	Plasma membrane
*MtCIPK13*	MtrunA17_Chr3g0091411	*MsCIPK11*	Chr3: 12692385–12693326	9.27	30,232.13	257	Plasma membrane
*MtCIPK14*	MtrunA17_Chr3g0091421	*MsCIPK11*	Chr3: 12693793–12694524	8.93	52,506.01	457	Plasma membrane
*MtCIPK15*	MtrunA17_Chr3g0107011	*MsCIPK9*	Chr3: 30606080–30609128	9.05	52,009.48	447	Plasma membrane
*MtCIPK16*	MtrunA17_Chr3g0107051	*MsCIPK10*	Chr3: 30640386–30641717	9.22	47,462.77	420	Plasma membrane
*MtCIPK17*	MtrunA17_Chr3g0114511	*MsCIPK11*	Chr3: 35974723–35976525	8.67	50,488.22	446	Plasma membrane
*MtCIPK18*	MtrunA17_Chr4g0001331	*MsCIPK12*	Chr4: 900468–902885	9.27	28,716.18	243	Plasma membrane
*MtCIPK19*	MtrunA17_Chr4g0013131	*MsCIPK14*	Chr4: 10571450–10575122	9.1	49,421.81	435	Plasma membrane
*MtCIPK20*	MtrunA17_Chr4g0063881	*MsCIPK24*	Chr4: 55303349–55314602	8.53	49,753.07	436	Plasma membrane
*MtCIPK21*	MtrunA17_Chr4g0069301	*MsCIPK18*	Chr4: 59217861–59224659	8.85	51,572.49	460	Plasma membrane
*MtCIPK22*	MtrunA17_Chr4g0072831	*MsCIPK24*	Chr4: 61783649–61786108	9.05	27,643.06	237	Cytoplasmic
*MtCIPK23*	MtrunA17_Chr4g0074111	*MsCIPK26*	Chr4: 62835031–62837115	9.34	25,004.1	214	Plasma membrane
*MtCIPK24*	MtrunA17_Chr5g0428031	*MsCIPK15*	Chr5: 29699497–29710190	9.26	50,104.11	444	Plasma membrane
*MtCIPK25*	MtrunA17_Chr5g0432471	*MsCIPK16*	Chr5: 33183949–33187482	8.97	50,856.92	446	Plasma membrane
*MtCIPK26*	MtrunA17_Chr5g0432491	*MsCIPK17*	Chr5: 33195433–33198761	6.82	50,243.62	441	Plasma membrane
*MtCIPK27*	MtrunA17_Chr5g0441011	*MsCIPK18*	Chr5: 39527588–39534318	6.72	50,665.32	446	Plasma membrane
*MtCIPK28*	MtrunA17_Chr7g0244521	*MsCIPK19*	Chr7: 33731410–33735051	8.92	52,017.29	453	Plasma membrane
*MtCIPK29*	MtrunA17_Chr7g0244561	*MsCIPK20*	Chr7: 33768481–33770580	5.69	52,186.74	465	Plasma membrane
*MtCIPK30*	MtrunA17_Chr8g0346071	*MsCIPK21*	Chr8: 9057047–9061702	8.58	53,028.89	465	Plasma membrane
*MtCIPK31*	MtrunA17_Chr8g0346081	*MsCIPK21*	Chr8: 9065393–9070388	9.22	50,033.79	443	Plasma membrane
*MtCIPK32*	MtrunA17_Chr8g0379091	*MsCIPK7*	Chr8: 39906567–39908530	8.93	52,494.6	465	Plasma membrane

## Data Availability

All data in the present study are available in the public database as referred in the Material and Method part.

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
