# Peer review of "Identification and Characterization of Abiotic Stress Responsive CBL-CIPK Family Genes in Medicago"

_ijms, 2021, doi:10.3390/ijms22094634_

Round 1
Reviewer 1 Report
Overall, the manuscript reads as one that is a survey of genes without any opportunity to ascribe function to their protein products. While the evolution of these families is interesting as presented, one is left with little depth of understanding beyond the in silico analyses of these genes. While expression profiles are suggestive, there is little to no biology associated with these analyses and the manuscript ends up being a speculative demonstration that indeed these are CBL's and CIPK's. Of course focusing on one or more members to do biology and demonstrate roles would be difficult partly due to the sheer number of candidate genes but the survey nature of the manuscript leaves it overall speculative. Specific comments are as follows:
1.) Line 53: genes do not function. Rather it's the proteins they encode. Please revise. I also feel that the authors are oversimplifying the process they are studying. For example in line 55 they state that AtCIPK8 “alone” but it is almost certain that this protein functions as part of a complicated process and overexpression is overcoming some limitation of the entire process. The authors should take care in how they are presenting these models.
2.) In line 91, the authors state that they found their candidates by annotation. It is surprising that homology searches also were not part of their selection approach, which makes one wonder if the statement in line 96 is fully inclusive.
3.) Line 103- predictive approaches do not “show” anything. They suggest. Please revise.
4.) It is not clear how one can claim “may be unique in Medicago” when the only comparison done was with Arabidopsis.
Author Response
Point 1: Line 53: genes do not function. Rather it's the proteins they encode. Please revise. I also feel that the authors are oversimplifying the process they are studying. For example in line 55 they state that AtCIPK8 “alone” but it is almost certain that this protein functions as part of a complicated process and overexpression is overcoming some limitation of the entire process. The authors should take care in how they are presenting these models.
Response 1: We thank the reviewer for pointing out this mistake, according to the reviewer' s suggestion, we revised this part in the revised version.
Point 2: In line 91, the authors state that they found their candidates by annotation. It is surprising that homology searches also were not part of their selection approach, which makes one wonder if the statement in line 96 is fully inclusive.
Response 2: We thank the reviewer for pointing out this issue. We did identified this genes by homology searching, but we did not interpret this part correctly. According to the reviewer' s suggestion, we revised this part in the revised version.
Point 3: Line 103- predictive approaches do not “show” anything. They suggest. Please revise.
Response 3: According to the reviewer' s suggestion, we revised this word in the revised version.
Point 4: It is not clear how one can claim “may be unique in Medicago” when the only comparison done was with Arabidopsis.
Response 4: In order to avoid potential misunderstanding, we deleted this sentence in the revised version.
Reviewer 2 Report
The authors have nicely present the current data on characterization of abiotic stress responsive CBL-CIPK family genes in two species of Medicago. The article is complex and it has many good features including the experimental design, accuracy and a deep focus on the data currently published in the literature, stating the key role of those two family genes in general response of plants to abiotic stress.
Author Response
We appreciate the reviewer for the positive evaluation of this manuscript.